# Sulfonated Block Copolymers: Synthesis, Chemical Modification, Self-Assembly Morphologies, and Recent Applications

**DOI:** 10.3390/polym14235081

**Published:** 2022-11-23

**Authors:** Claudia I. Piñón-Balderrama, César Leyva-Porras, Alain Salvador Conejo-Dávila, Erasto Armando Zaragoza-Contreras

**Affiliations:** Centro de Investigación en Materiales Avanzados, S.C. Miguel de Cervantes No. 120, Complejo Industrial Chihuahua, Chihuahua 31136, Mexico

**Keywords:** sulfonated block copolymers, phase-transition, polymer morphology, structure–property relationship

## Abstract

Scientific research based on the self-assembly behavior of block copolymers (BCs) comprising charged-neutral segments has emerged as a novel strategy mainly looking for the optimization of efficiency in the generation and storage of electrical energy. The sulfonation reaction re- presents one of the most commonly employed methodologies by scientific investigations to reach the desired amphiphilic character, leading to enough ion concentration to modify and control the entire self-assembly behavior of the BCs. Recently, several works have studied and exploited these changes, inducing improvement on the mechanical properties, ionic conduction capabilities, colloidal solubility, interface activity, and stabilization of dispersed particles, among others. This review aims to present a description of recent works focused on obtaining amphiphilic block copolymers, specifically those that were synthesized by a living/controlled polymerization method and that have introduced the amphiphilic character by the sulfonation of one of the segments. Additionally, relevant works that have evidenced morphological and/or structural changes regarding the pristine BC as a result of the chemical modification are discussed. Finally, several emerging practical applications are analyzed to highlight the main drawbacks and challenges that should be addressed to overcome the development and understanding of these complex systems.

## 1. Introduction

A block copolymer (BC) is defined as a macromolecule composed of at least two different homopolymers organized in sequences or blocks of different chemical composition. Unlike a random copolymer, the main difference is constitutional, because a block only comprises a succession of chemically identical monomer units. Additionally, a BC is thermodynamically different to a simple physical blend of homopolymers due to the existence of a covalent chemical bonding between the different polymer segments. This chemical bond plays an important role in the BC’s structure, because it is responsible for constraining the macroscopic-scale phase separation behavior typically found in physical blends of homopolymers [1,2,3]. As a consequence of the chemical compositions of the blocks and immiscibility of the consitituents blocks, a phase separation at microscopic scale occurs, which is usually named as microphase separation [4]. Because of the process of microphase segregation, the BCs tend to self-assembly into domains with periodicities varying in the order of the nanometers (≈1–100 nm). This provides a potential for the development of nanomanufacturing technologies such as bottom-up or top-down at competitive costs and ease of implementation. In addition, scalable procedures, such as applications involving tailor templates and spatially controlled systems in the field of the electronic devices [5,6], production and storage of clean energy [7,8,9,10], and drug delivery systems [11,12,13,14], are possible. 

The self-consistent field theory (SCFT) explains the ordering and phase separation behavior of block copolymers in terms of the incompatibility of two distinct monomers by the Flory–Huggins interaction parameter (χ), the degree of polymerization (*N*), and the volume fraction (*f*) [15,16,17,18,19]. The solutions of the equations proposed in the theory, compare the free energies for different phases and summarize the results in a well-known phase diagram (Figure 1). The parameters (*N*, *f*, χ) can be controlled during the synthesis processes—the polymerization degree and the volume fraction by the stoichiometry of the reaction, and the interaction parameter by the correct selection of the monomers used during the polymerization. The predicted structures reported theoretically and experimentally range from lamellar, hexagonal, cylindrical, and gyroid arrangements regarding variatons of the parameters *N*, *f*, and χ. 

Amphiphilic block copolymers (ABCs) represent a class of functional block copolymers (BCs) that compromise segments with different chemical composition, polarity, and cohesion energy. Generally, ABCs are composed of at least one segment of hydrophilic nature and another with a hydrophobic character. The growing interest in ABCs lies in the advantages that are obtained from the hybrid materials to form nanostructures with unique properties. This may include applications that require colloidal solubility, interface activity, stabilization of dispersed particles, and ionic conductivity for application such as fuel and solar cells [12], drug delivery systems [11,13], and nanomanufacturing of electronic devices [20,21,22,23]. 

One of the most-employed techniques for obtaining ABCs is by the introduction of a polar functional group into one segment of the BC. A common methodology for obtaining ABCs is sulfonation, which is a chemical reaction for introducing the hydrophilic character of the BC. Sulfonation is a substitution or additional chemical reaction used to incorporate sulfonic acid groups (SO_3_H) in a polymer chain by a chemical bond to a carbon atom or less commonly to a nitrogen atom [24]. Chemical compounds such as sulfuric acid, sulfur trioxide, and their derivatives, such as acetyl or alkyl sulfates and chlorosulfonic acid, are commonly employed as sulfonant agents (SA). The sulfonation is carried out in two different ways. The first is a post-modification reaction, in which, the BC is synthesized, and then the polar character in one of the segments is chemically introduced, and the second is the synthesis of the BC in which one polar monomer and one non-polar monomer are involved. From these two approaches, the post-modification is preferred because it allows prediction of the types and quantities of ionic moieties, at the time that the reaction conditions are more easily implemented. Polystyrene has been widely explored as the segment for sulfonation because the reaction proceeds easily through an electrophilic aromatic substitution. However, other polymers are also being studied, including aromatic rings [25,26] and polydienes previously subjected to a hydrogenation, for a selective sulfonation. Once the incorporation of amphiphilic character has been introduced, it is expected that original predicted morphology and structure may change due to different chemical interactions between the polar groups such as electrostatic [27], dipole–dipole [28,29,30], Van der Walls interactions [31,32], or hydrogen bonds [33]. 

This review aims to present a description of recent works focused on obtaining ABCs, specifically, those that have introduced the amphiphilic character by the sulfonation of one of the segments, and additionally those that have evidenced the morphological and structural changes induced regarding the pristine BC. For this purpose, first, a brief introduction of some synthetic techniques available for obtaining BCs is included. Then, the sulfonation methodology for imparting the amphiphilic character in the BC, either through a chemical post-modification, or by the use of sulfonated monomers, will be presented. The third section addresses the modifications of the structure and morphology, evidenced by characterization techniques. Finally, some applications that take advantage of the self-assembly the BCs, and how the amphiphilic character modifies the ordering process of the unmodified BC are discussed.

## 2. Phase Diagram of BCs

One of the most important features of BC is also very interesting, because the point of view of the application is the ability to self-assemble into different ordered nanostructures via the process of microphase separation. Microphase separation is a thermodynamical process driven by the enthalpy of demixing. This enthalpy is proportional to the χ value, being inversely proportional to the temperature. The larger and more positive χ interaction parameter value, the greater the repulsive energies between different chains, and consequently, the segregation strength becomes more pronunced. Additionaly, the enthalpic penalization of chain stretching depends on the polymerization degree (N or length of the chains). In the phase diagram (Figure 1), the x-axis represents the volume fraction of each block (from 0 to 100%) and the y-axis represents the χN product. The diagram express a balance between enthalpy and entropy employed to parameterize the block copolymer phase behavior, along with the composition of the block copolymer [34]. The phase diagram corresponds to a conformationally symmetric diblock copolymer, calculated with the SCFT. The regions of the stability of disordered, lamellar, gyroid, hexagonal, body cubic centered, and closed-packed spherical phases, are indicated. The phase diagram constructed from experiments of a diblock based on a poly(styrene-b-isoprene) system could be employed as a guide for the obtainment of a desired nanostructure fixing the experimental conditions that gives rise to a specific self-assembled, ordered system. For example, in the BC of symmetric composition, it is expected that the blocks self-assemble into alternating layers (lamellar (L)) if the degree of incompatibility is high enough to overcome the weak segregation limit (WSL, χN ≈ 10). For more asymmetric copoplymers, the phases with curved interfaces such as hexagonal packed cylinders (H;Q230), double gyroid-phase (GYR; Q229), or body-centered cubic (CPS) phases, becomes more energetically favourable. In the strong segregation limit (SSL, Χn ≈ 12), the following sequence is observed for a diblock copolymer system polystyrene-polyisoprene (PS-PI): f PS > 0.77, CPS, 0.66 > f PS < 0.77, Q230, 0.62 > f PS < 0.66, Q229, 0.62 > f PS < 0.34, L, and their corresponding phase inversion. 

## 3. Techniques for Obtaining BCs

The development of the synthesis of BCs began in the 1950s with the discovery of free-termination anionic polymerization [37]. This methodology was named “living polymerization” and it made possible the sequential addition of different monomers to various carbanion-terminated linear polymer chains. The term “living” refers to the polymer chain that has not undergone termination reaction. Hence, the end of the first polymer chain could be reactivated and reinitiate to increase the molecular weight of the existent or to react with a different monomer to produce a copolymer. Once the first monomer has been completely consumed, upon the addition of a second monomer, the active centers are able to continue the polymerization through the addition of a second monomer. Many methods of living polymerization have been studied, including anionic, cationic, ring-opening, and group transfer reactions. However, some of them require demanding reaction conditions, such as low tolerance to impurities, very low humidity, and strict temperature of polymerization. Later, with the development of controlled radical polymerization (CRP), the living polymerization reaction emerged as a more efficient and simple option for polymer synthesis. The understanding and evolution of the CRP allowed for the combination of the benefits of both techniques, thus giving rise to the “living controlled polymerization methods”. The implementation of this novel method represents a new available tool for polymer chemistry that offers great chances for more sustainable routes to obtain tailor-made polymers with high precision on polymerization process, in which the versatility of conventional radical polymerization could be retained. The living/controlled radical polymerization techniques that have the potential to revolutionize a large part of the polymer industryand are, therefore, receiving attention are atom-transfer radical polymerization (ATRP), nitroxide-mediated polymerization (NMP), and reversible addition–fragmentation chain transfer (RAFT). The mechanisms of living polymerizations employed as active center radicals or organometallic species and are based on the evolution of dynamic equilibrium between active and dormant species (species that are still “alive” but with a much lower reactivity). In order to reach a good control on the polymerization, the following aspects should be considered:(1)The initiation stage must be much faster than the propagation because it needs the chains to start and grow simultaneously.(2)The concentration of propagating radicals is very low to substantially decrease the termination events, whereas the chains are growing with very few side events like termination or chain transfer reactions.(3)A rapid exchange between dormant and active species is also required, favoring most of the growing chains that remain in the “dormant” state and only a small fraction of active radicals as end groups in the chain.

The use of living control polymerization techniques not only provides good control on the molecular weights but also functionalities and topologies capable of self-assembly into well-defined supramolecular structures. The simplest case of BCs is of AB type, which is the polymerization of two distinct monomers (e.g., A styrene and B butadiene); however, a three- or multiple-step polymerization reaction is also possible for the preparation of ABA, BAB, and a wide variety of sequences and architectures. This may result in promising materials for applications in which well-defined, targeted nanostructures may improve the performance of devices such as photonic crystals [38], high-density information storage devices [39], templates for metals [40], and semiconducting nanowires [41].

A more detailed explanation of the different mechanisms of controlled polymerization techniques are out of the scope of this review; for this purpose, please refer to the following extensive bibliography [42,43,44,45,46]. In this work, several works that made use of BC synthesized by a controlled living polymerization mechanisms will be reviewed. 

## 4. Amphiphilic Character of BCs through the Sulfonation

The amphiphilic character in a BC can be introduced by two strategies. First, the BC is synthesized and then chemically modified to introduce polarity to one of the segments; this is called a post-modification reaction. The second strategy is introduced when one polar monomer and one non-polar monomer are employed for the BC synthesis. The post-modification reaction is commonly reported in the literature because it allows us to predetermine types and quantities of ionic moieties, while the reaction conditions are more easy to be implemented. 

Amphiphilicity can be carried out by adding acid-tethered functionalities to a segment of the BC; for example, by introducing sulfonic or phosphonic acid groups, or functionalization with positive charged moieties such as quaternary ammonium or phosphonium groups. Among the available techniques for introducing the amphiphilic character, sulfonation has been widely explored because the versatility of the technique employing several sulfonating agents, reaction mediums, and conditions (mild, strong). The sulfonation of the BCs provides the material amphiphilic properties, which induces phase behavior due to block polarity differences. For this, sulfonation selectivity is needed; i.e., only one of the blocks is modified to facilitate phase segregation. 

This section describes the sulfonation of block copolymers synthesized by strategies that purely involve living polymerization mechanisms of vinyl monomers or their combination with other mechanisms. The information has been organized both chronologically and in accord with the sulfonating agent used, so that the results can be analyzed in a logical and sequential manner. 

Polymer sulfonation has been performed since the early 1950s. According to the literature, the first works were reported by Reynolds and Kenyon [47] and by Roth [48], but the sulfonation of block copolymers was not reported until a few decades later. The “acetyl sulfate (acetoxysulfonic acid)” complex, obtained from acetic anhydride and concentrated sulfuric acid, is the most used source for the selective sulfonation of block copolymers. This complex was reported in the early 1980s by Zhou and Eisenberg [49] for the sulfonation of cis-1,4-polyisoprene, and by Weiss et al. [50] for sulfonating polystyrene. Concerning block copolymers, Storey et al. used for the first time the acetyl sulfate complex in dichloromethane-hexanes solution to modify three-arm star-branched block copolymer ionomers, consisting of butadiene elastomeric inner blocks and oligostyrene ionic outer blocks. Neutralization of sulfonic acid with a base in THF followed the sulfonation of the styrene oligomers. With this method, a sulfonation degree close to 90% was achieved. Among other results, it was found that the mechanical behavior was affected by the counterion used in the neutralization [51]. Later, Erdoğan et al. partially sulfonated poly(styrene-b-methyl methacrylate) copolymers to convert them into ionomers for application in proton exchange membranes. The copolymers were obtained via sequential copper-mediated atom transfer radical polymerization, with blocks of 7 and 12% poly(methyl methacrylate). Sulfonation degree of polystyrene block was reported in the range of 20 to 30%. The copolymer characterization was carried out by FTIR, which showed the characteristic signals of the SO_3_- group (1005 and 1032 cm^−1^); furthermore, it was found that the glass transition temperature increases with the sulfonation degree [52]. In addition, the acetyl sulfate complex was used by Tsang et al. for the sulfonation of poly([vinylidene difluoride-cohexafluoropropylene]-b-styrene) for application in proton exchange membranes. The sulfonation degree of the polystyrene block was also varied to correlate with the ion exchange capacity. For the synthesis of the diblock, a two-step process was used, the first to obtain the fluorinated block by emulsion polymerization using chloroform (Cl_3_CH) as a chain transfer agent. Subsequently, the CCl_3_-terminated block was used as a macroinitiator for the propagation of the polystyrene block via ATRP. Finally, acetyl sulfate was used to selectively and partially postsulfonate only the polystyrene block. The size of the membrane ionic aggregates was found to increase with the sulfonation degree [53]. In addition, Huang et al. selectively modified block copolymers polystyrene-b-(1,3-cyclohexadiene), which was obtained by anionic polymerization initiated with sec-butyllithium. Specifically, the polycyclohexadiene block was modified by fluorination, while the polystyrene block by sulfonation, using the acetic anhydride-sulfuric acid system in 1,2-dichloroethane. The sulfonation was characterized by FTIR (1260, 1150, and 1065 cm^−1^) and quantified by ^1^H NMR through the analysis of the protons of the aromatic ring. An apparent interaction of the SO_3_- groups with the counterions (Cs^+^) of a step of the polycyclohexadiene modification (hydrogenation) prevented an adequate quantification of the sulfonation [54]. Afterward, Erdoğan et al. synthesized a series of polystyrene-b-poly(2,2,3,3,3-pentafluoropropyl methacrylate) multiarm copolymeric ionomers with a crosslinked divinylbenzene core, by using a synthesis sequence via a combination of atom transfer radical polymerization (ATRP) and Diels–Alder click reaction. The postsulfonation of the internal polystyrene segments was carried out with the complex at 20 °C. The presence of peaks at 1006 and 1033 cm^−1^ in the FTIR spectrum evidenced the presence of sulfonate groups. Additionally, ^1^H NMR showed a new peak at 7.5 ppm giving evidence of the sulfonation. The membranes obtained from the sulfonated multiarm copolymers and sulfonated poly(phenylene oxide) showed an ion exchange capacity dependent on the sulfonation time [55]. Finally, Ruiz–Colón et al. elaborated membranes for fuel cells from a block of poly(styrene-isobutylene-styrene), obtained by a Friedel–Craft reaction, to which poly(vinyl phosphonic acid) was grafted by ATRP. The post sulfonation of the styrene block was carried out with the acetyl sulfate complex at 0 °C. The sulfonation was characterized by elemental analysis through the sulfur content. The sulfonation degrees obtained were close to 20%, less than expected, which was attributed to a high solubility of the poly(vinyl phosphonic acid) block. The membrane showed high proton conductivity at 80 °C and a lower permeability to methanol compared to Nafion. These improvements were related to the ionic interaction of the phosphonate and sulfonate groups [56].

The sulfur trioxide-triethyl phosphate complex is the second-most reported source of SO_3_- for the modification of BCs. The first reports of this complex for sulfonation date back to the mid-1970s, where Noshay and Robeson [57] and Rahrig et al. [58] used this complex to sulfonate polysulfone and polypentenamer, respectively. As for block copolymer modification, Valint and Bock [59] synthesized block copolymers by anionic polymerization, using butyl lithium as initiator, to explore hydrophobically associative polymers to provide viscosification: the tert-butylstyrene and styrene base copolymer, the former of which was used to constitute the hydrophobic block and the latter, to a greater extent, as the hydrophilic block after selective post-sulfonation. This group used for the first time this complex. In addition to block copolymers, the random copolymer and homopolymer of styrene were also modified. In all cases, the sulfonation method showed efficiencies above 80%, although the viscosification effect showed greater efficiency in the block structure. The group of Gatsouli et al. also used this complex for the modification of poly(sulfonated styrene-b-tert-butylstyrene). Micelles of this copolymer were used for the synthesis of CdS and CdSe nanoparticles. The unsulfonated copolymer was synthesized by anionic polymerization initiated by sec-butyllithium in benzene solution. The sulfonation of both copolymers was characterized by FTIR, in which the characteristic peaks of the sulfonate group were observed, whereas the degree of sulfonation was characterized by titration with acid, giving values close to 100%. During the synthesis of the nanoparticles, the sulfonate groups bond with the Cd^2+^ groups, inducing the formation of micelles [60]. Additionally, Huang et al. obtained a series of poly(1,3-cyclohexadiene) star copolymers through convergent anion polymerization, using (chlorodimethylsilyl)styrene as the core. Once the copolymers were synthesized, the cyclohexadiene ring was exposed to several modification processes, including a sulfonation degree of 70%. Thermogravimetric analysis showed that sulfonation provides the copolymer increased stability [61]. More recently, Politakos et al. synthesized a polystyrene-b-polyisoprene copolymer by sequential anion polymerization. Subsequently, this polyisoprene block was partially sulfonated to produce poly(sulfonated isoprene-co-isoprene), giving the copolymer amphiphilic character. Selective sulfonation was carried out, leaving the polystyrene block intact. The appearance of the signals at 1042 and 1184 cm^−1^ and 3.75 ppm in FTIR and ^1^H NMR, respectively, gave evidence of the incorporation of the sulfonate groups in the polyisoprene. In addition, the ^1^H NMR, allowed us to calculate a degree of sulfonation close to 43% [62].

In addition, the sulfonation of block copolymers has been carried out with chlorosulfonic acid. According to the literature, chlorosulfonic acid was one of the first polymer sulfonating agents [48]. Concerning block copolymers, Liang and Ying [63] reported poly(styrene-butadiene-4-vinylpyridine) copolymers by sequential anionic polymerization with n-butyllithium as the initiator and benzene as the solvent. After the synthesis of the copolymer, the polystyrene segments were sulfonated and then neutralized with a treatment in sodium hydroxide. The resulting membrane had substantial cation and anion exchange capabilities. Later, Xu et al. [64] reported the synthesis of sulfonated poly(styrene-b-vinylidene fluoride-b-styrene) triblocks, in which the degree of post-sulfonation was varied. The triblock synthesis involved an ATRP-mediated step to extend a polystyrene block, followed by post-sulfonation. The sulfonation was analyzed by FTIR and ^1^H NMR, the latter resulting in a modification of the order of 10 to 50%. The advantage of the block copolymer structured membranes was reflected in the ionic conductivity, compared to the random copolymer.

In addition to the sulfonating agents mentioned above, other methods have also been used; for example, Leemans et al. [65] resorted to living anion polymerization, using sec-butyllithium as initiator, to synthesize a methyl methacrylate-glycidyl methacrylate block copolymer, which was selectively modified from the oxirane ring opening of the glycidyl methacrylate block, to obtain hydroxysulfonate groups. Thanks to such modification, the block copolymer showed surface active properties (critical micelle concentration of 0.5 g/dL). A phase transfer catalyst was used to perform the nucleophilic substitution, in which the copolymer was suspended in an aqueous solution of tetrabutylammonium bromide and anhydrous sodium sulfite. After a reaction period of 30 h at 80 °C, a sulfonation of 1.9 mmol SO_3_- g^−1^ was obtained by titration with the base. Yang and Mays synthesized linear and star copolymers of polystyrene and poly(4-tert-butylstyrene), using sec-butyllithium-initiated anionic polymerization techniques. The sulfonation of the PS blocks was carried out by using the sulfur trioxide-triethyl phosphate complex in 1,2-dichloroethane. In the final stage, the sulfonic acid was neutralized with sodium methoxide. This method allowed the selective sulfonation of polystyrene blocks with sulfonation degree greater than 80%; the highest values were obtained with linear copolymers. It is worth mentioning that the star structure was also less soluble in aqueous phase than the linear structures [66]. Likewise, Li et al. synthesized copolymers with a fluorinated acrylic base block and a polystyrene block by using the ATRP technique. The polystyrene block was partially sulfonated by using sodium lauroyl sulfate (obtained from the reaction of chlorosulfonic acid and lauric acid). The sulfonation was carried out in cyclohexane as the solvent at 50 °C. Sulfonation was determined by FTIR (absorptions at 1008 and 1032 cm^−1^) and elemental analysis, indicating sulfonation levels between 5 and 28% [67]. Finally, Khawas et al. reported a poly(3-hexylthiophene)-b-polystyrenesulfonic acid semiconductor block copolymer prepared by a Kumada catalyst transfer polymerization synthesis sequence, to obtain the bromobenzyl end functionalized poly(3-hexylthiophene) block, and atom transfer radical polymerization, to extend the polystyrene block. The sulfonation of the second block was carried out by exposing the copolymer to a solution of phosphorous pentoxide and concentrated sulfuric acid. Infrared spectroscopy confirmed the presence of SO_3_- groups (1168, 1125, and 1038 cm^−1^), whereas ^1^H NMR indicated almost 100% sulfonation of the polystyrene block. Additionally, the morphology of the copolymers was found to range from spherical to disc-like nanostructures and the ionic and grain boundary conductivity, determined by impedance spectroscopy, were 5.25 × 10^−4^ and 4.66 × 10^−6^ S cm^−1^, respectively [68]. 

As observed, the sulfonation of block copolymers has been carried out mainly by means of the acetyl sulfate complex, obtained from the reaction between acetic anhydride and concentrated sulfuric acid in 1,2 dichloroethane. Polystyrene was the most common block found in these studies, because the aromatic ring can be selectively modified. Table 1 and Table 2 report the more relevant works carried out in the last decade. The BCs were synthesized by the control living polymerization strategy, in which the sulfonation was carried out either by employing a sulfonated monomer or by a post-chemical modification of the existing blocks. 

## 5. Morphological and Structural Changes Induced by the Sulfonation

Controlled supramolecular self-assembly of BCs has being explored, leading to extraordinary materials that are useful in a wide variety of applications. The BCs possess the ability to self-assemble into well-defined nanostructures with controllable size, shape, and periodicities of domains. This process employs the chemical dissimilarity between the component blocks as a driving force. A wide variety of desired morphologies may be obtained by variations in the polymerization degree (*N*), the volume fraction (*f*) of each block and the interaction parameter (χ) [4,17,19,90]. The three parameters (χ, *N*, *f*) could be experimentally controlled during the synthesis processes, the polymerization degree, and the volume fraction by the stoichiometry of the reaction, whereas the interaction parameter by the selection of the monomer pair used for the polymerization reaction. Resulting nanostructures and morphologies may be predicted for neutral BC, and are widely reported theoretically and experimentally by the literature ranging from lamellar, hexagonal, cylindrical, and gyroid arrangement even in bulk, solution, or in the form of thin films [91,92]. However, when a polar functionality or charged-neutral segments are employed for the synthesis, the amphiphilic character is induced into the BC. The self-assembly process exhibited enormous differences due to the different physical interactions between decharged segments, so the morphology and structure may not be easily predicted. In fact, recent works had shown considerable inconsistences with the existing theory evidencing that phase separation between neutral-charged is much more complex [84,93,94,95,96,97]. Despite the extensive research in this field, a fundamental understanding of microphase separation of charged BC needs to be more thoroughly developed. Therefore, herein it is pretended to highlight those changes. This section has been focus to review relevant works that compromise the relationship of structure and properties of block copolymers with amphiphilic character introduced through sulfonation. 

### 5.1. Phase Behavior of Sulfonated Block-Copolymers

The phase behavior of BC is a thermodynamically favored process that, as has been mentioned before in previous sections, is caused by chemical dissimilarity between blocks and low affinity of the chemically distinct constituent blocks.

In sulfonated systems, the segregation strength of the blocks is increased because the copolymer is composed of hydrophilic and hydrophobic segments with lower miscibility between each other in comparison with neutral constituent blocks. In fact, it has been reported that the interaction parameter increases due to the increase in repulsive energies between dissimilar chains [33].

The phase behavior of sulfonated block copolymers and its impact on the mechanical properties has been extensively reported. Pioneer studies in this area started with the evaluation of the physical properties of an ionomer (polymer where the segment containing the ionic moiety does not constitute more than 15 mole percent), polyurethanes-hydroxybutyl terminated, poly(chloropropylmethyl-dimethylsiloxane) and poly(tetramethylene oxide). The materials were chain-extended with N-methyldiethanolamine (MDEA) and ionized. The authors studied the effect of mixing soft segments, chain extenders, and zwitterionization on the extent of phase separation and its physical properties. The resulting ionic aggregation did not affect the assembled morphology, but it had a positive impact on the tensile and viscoelastic properties of the BCs [98]. Wang et al. [99] studied a multi-block copolymer system based on poly(arylene ether sulfone) with selectively cross-linked hydrophobic domains to produce bulk materials. The sulfonated BCs were cast to produced membranes, and its performance against temperature, stress, and humidity was evaluated. The sulfonated cross-linked membranes showed excellent thermal, mechanical, and oxidative stability from the high cross-link efficiency with a gel content of over 90%. Additionally, the swelling ratio of the BCs was observed to be suppressed to 25% at 25 °C in the densely cross-linked hydrophobic blocks, which resulted in the improvement of the proton conductivity and dimensional stability. Tashibana et al. [100] have investigated ion gel membranes composed of ionic liquids (ILs) and sulfonated polyimides (SPIs). Different sequential distribution of the ionic groups was studied in random and multi-block copolymers. The materials were prepared aiming to obtain a polymeric membrane useful for the separation of CO_2_. The authors observed that a multi-block copolymer exhibited a clearer phase-separated structure evidenced by small angle X-ray scattering (SAXS) profiles; however, the strain at the break of the multi-block copolymer was lower owing to the brittleness of the non-ionic phase. A strategy for incorporating fluorinated groups was implemented to improve the CO_2_ permeability and the ductile properties, resulting in a high efficiency of the gas separation. Ths authors concluded that the polyimide sequence, in addition to the chemical structure of monomers, affects the mechanical properties of the sulfonated polyimide/ionic liquid composite membranes and its capacity for the separation of the greenhouse gas. Yang et al. [101] studied the proton conductivity of poly(ether sulfone) multi-block copolymers grafted with densely pendant sulfoalkoxyl side chains for proton exchange membranes. The authors reported that the densely grafted side chains enhanced the phase separation process and the formation of broader hydrophilic channels that cause a positive impact on the performance as a proton-conductive membrane. Loveday et al. [102] carried out structural investigations of BC, incorporating ionic species in the chain of butadiene-tert-butyl methacrylate and butadiene/styrene-tert-butyl methacrylate BCs, and converted into the ionomer form through the sulfonation reaction. The observed morphologies showed a transformation from being non-oriented and rod-like to spheroids promoted by the interaction of ionic domains. Furthermore, an alteration in the glass transition temperature and a relationship with the length of the ionic segment. Advanced electrical materials, with desirable mechanical strength, and a low glass transition temperature exhibiting high ionic conductivity endowed by the nature of ionic block copolymers, have been reported [95].

### 5.2. Nanoscale Morphologies Self-Assembly of Sulfonated Block-Copolymers

The self-assembly process into ordered nanostructures such as lamellar, gyroid, hexagonal packed cylinders and body cubic-centered phases could be attained when BC are subjected to some kind of annealing. The most commonly reported by the literature are the solvent vapor annealing (SVA) [103,104,105] and the thermal annealing (TA) [106,107]. The SVA consists of the exposition of the BC to a vapor of a solvent or mix of solvents, taking into account the solubility parameter of the solvent and the components of the BC, aiming that swelling occurs preferentially in one of the segments. TA is a thermal treatment that promoted the formation of ordered morphologies because polymeric chains are provided with enough energy and mobility to reach an equilibrium configuration. For this to occur, it is necessary to heat the BC well above the glass transition temperature (Tg) of each component block in the system.

A wide variety of works have revealed that the self-assembly of nanostructured BC thermally or solvent vapor annealed treated, exhibit higher segregation strength and better performance. For example, ionic conduction capabilities, in comparison with random copolymers since the transporting of H^+^ ions is optimized by conductive pathways, better defined by the order of ionic segments [7,108,109]. Controlling the conductivity of a BC, by the modification of the nanostructure size, shape, and periodicity of the ionic domain, is another interesting research area finding fields of application in the catalysis [110], gas separation media [111,112], and their use as polyelectrolytes in the proton exchange membrane fuel cell, and battery industry [10,113,114,115]. Rubatat et al. [116] studied the structure−properties relationship in proton-conductive sulfonated polystyrene−polymethyl methacrylate BCs. Several degrees of sulfonation were studied and the evaluations of the self-assembled morphologies were carried out. They observed that the ionic conductivity, normalized by the volume fraction (*f*) of the conductive domains formed by PS, sPS, and water content, increased monotonically with the content of sulfonic acid groups, and with the distinct morphologies: isotropic phase < cylindrical hexagonal phase < hexagonally perforated lamellar phase < lamellar phase. A similar work was reported by Piñón–Balderrama et al. [33] for polymethyl methacrylate and polystyrene BC system. The BCs were converted into their amphiphilic form, introducing sulfonic acid groups into the aromatic rings contained in the PS segment. Because of the chemical modification, it was observed that the self-assembly behavior of the entire BC was significantly modified. Once equilibrium morphologies were obtained, after exposition to solvent vapor (SVA) or thermal annealing (TA), the lamellar nanostructures observed in the pristine BCs become micelle, worm-like, and hexagonal packing, evidenced by atomic force microscopy of the thin films. According to the authors, the differences in the process of the microphase separation were attributed to the introduction of sulfonic acid groups that form hydrogen bonds between the charged segments, modifying the chemical interactions between the polymer chains, inducing an increase in the interaction parameter value. Mineart et al. [117] prepared a midblock sulfonated multiblock copolymer, aiming to obtain high water and ion transport. They investigated the swelling kinetics and temperature dependence of water uptake of the elastomeric thermoplastic BC. The self-assembly process was observed upon the presence of a polar liquid. Small-angle X-ray scattering and gravimetric measurements allowed us to establish a correlation of nanostructural changes with macroscopic swelling to establish a fundamental structure–property behavior. Recently, Politakos et al. [62] published an approach for obtaining various nanostructures. They employed a diblock based on poly(styrene-b-isoprene) (PS-b-PI) and made a comparative study with its hydrogenated and sulfonated derivatives. The hydrogenation of PI showed an enhanced immiscibility and greater segregation strength by the increase of χ, whereas the sulfonation introduced an amphiphilic character. Well-ordered, hexagonally close-packed cylinders were observed for the hydrogenated samples. In contrast, the partially sulfonated samples showed the formation of horizontal cylinders cast with cyclohexane as the solvent. Upon exposition at 80 °C, the morphology was altered to a possible micellar structure. Similar results were observed when toluene was employed as the casting solvent, and the thermal annealing was carried out at 120 °C. Qiang et al. [118] published a novel methodology for obtaining nanometer and micrometer-size Janus discs with controlled shape, size, and aspect ratio from the amphiphilic sulfonated triblock copolymer based on poly(styrene-b-butadiene-b-methyl methacrylate) (PS-*b*-PB-*b*-PMMA). The desired morphology combined a Shirasu porous glass with evaporation-induced confinement assembly to yield uniform prolate ellipsoidal microparticles with axially stacked lamella–lamella morphology. Subsequently, the authors take advantage of the amphiphilic nature of the sulfonated block to absorb gold nanoparticles to the surface of sulfonated polystyrene and probe the ability to stabilize Pickering emulsions. Sing et al. [93], have demonstrated theoretically that varying the charge of a block copolymer is a powerful mechanism to tune nanostructures and that highly asymmetric charge cohesion effects can induce the formation of nanostructures that are inaccessible to conventional uncharged block copolymers, including percolated phases desired for ion transport. The authors made use of hybrid self-consistent field theory and liquid state theory for accurately relating length scales and enabling the articulation of the thermodynamics of the BCs systems. With theoretical studies, they were able to construct two phase diagrams upon the introduction of charged segment. The created phase diagrams represent the volume fraction of the charged block in the x-axis and the immiscibility product of the χN on the y-axis. One diagram considers the influence of the charge and the counterion solubility but ignored the electrostatic cohesion, while the other diagram is based on the presence of Coulombic interactions. It was explained that with the presence of Coulombic interactions and soluble or unsoluble counterions, the phase separation behavior and hence the resulting nanostructure could be greatly modified. Gavrilov et al. [119] carried out a theorical investigation of the phase behavior of melts of block-copolymers with one charged block by dissipative particle dynamics in the presence of electrostatic interactions. The authors constructed a neutral copolymer by grafting counterions to the corresponding co-ions in the charged blocks. Both BC, charged and neutral, were compared in two different phase diagrams. It was found that the phase diagrams’ differences were subtle, and the same phases in the same order were observed regarding the conventional neutral phase diagram; however, for the theoretical calculations it is assumed that a χ value is equal to 0, which means that both blocks are fully compatible. Both diagrams are included in Figure 2. Furthermore, the study includes the order–disorder transitions of a lamellar structure depending on chain length. Here, they conclude that the presence of a counterion entropy prevent the formation of ordered structures, becoming more pronounced as the length of chains gets larger. Park and Balsara [120] evidenced that the phase diagram of ionic/neutral BC is much more complex than that in an uncharged system. They employed poly(styrenesulfonate-methylbutylene) (PSS-PMB) BCs and evaluated the phase morphology behavior by using sulfonation degrees of 17,24 and 38%. The compositions of the BCs were nearly symmetric; for a system for such a composition according to the self-consistent field theory, it would be expected to exhibit a lamellar morphology; however, upon the introduction of a polar segment, the segregation strength was modified and hence ordered in lamellar, gyroid, hexagonally perforated lamellae, and hexagonally packed cylinder phases were obtained for the charged PSS-PMB system. Additionally, the authors employed a variety of methodologies for predicting the Flory–Huggins interaction parameter, finding larger values of χ between the sulfonated and non-sulfonated blocks. At these conditions, ordered nanostructures were observed even at very low molecular weight block copolymers (1.8K–1.4K). It is worth mentioning that at low polymerization degrees, the systems would be expected to show a disordered phase behavior, however, as observed in Figure 3, TEM characterization showed the formation of gyroid, lamellar and hexagonal packed cylinders (Figure 3). With the goal of replacing commercially available perfluorosulfonic acid (Nafion) membranes but at lower cost and hydrogen permeability. Liu et al. [121] synthesized BC systems of sulfonated poly(arylene ether) with densely sulfonated segments containing mono, di, and tri-tetraphenylmethane. Small-angle X-ray scattering and transmission electron microscopy showed aggregation of ionic clusters. The sulfonated system exhibited spherical hydrophilic ionic clusters, indicating that considerable phase morphology was induced. In addition, it was observed that as the length of the hydrophilic segment increases, the size of the hydrophilic ion clusters also increases, and the dispersed ion channels were more effectively connected, favoring the conductivity properties of the proton exchange membrane materials.

## 6. Applications of ABCs

Initially, the research on block copolymers was based on the use of polystyrene and butadiene monomers, trying to combine rigid and flexible segments to maximize the mechanical properties. Later, passing to the use of micelles and the characterization of the molecular weight and morphology, continuing with the synthesis with polyethylene oxides, determining the self-assembly morphology, to the actual stage of the applications such as drug delivery [122]. Evidently, the evolution of block copolymers has led to the discovery of novel applications, including electronics, fuel cells, solar cells, and lithography. Thus, in this section the use of sulfonated BCs will be reviewed.

In general, the intrinsic properties of the polymer are modified with sulfonation. The Tg of the polysulfone increased by 130 °C upon addition of the sulfonante agent [57]. In hydrogenated sulfonated polypentenamers, Tg increased from −100 °C to 25 °C with the increase in the percentage of sulfonation from 1.9 to 17.6%, respectively [58]. The neutralization or replacement of the sulfonic groups by amine groups of different chain length, in sulfonated polystyrene, decreased the Tg and the melt viscosity. This is because amine groups reduce ion-dipole interactions between sulfonic groups [50]. In cis-1,4-polyisoprene, the cyclization of the sulfonated rubber prevents the Brownian motion of the polymer chains, which is observed as the increase in the elastic modulus [49]. In sulfonated poly(styrene-isobutylene-styrene) (SIBS) block copolymers, the elastic modulus (E) and hardness (H) increased up to a certain level of sulfonation of 64% in weight. A further increase in the sulfonation hinders any mechanical reinforcement [123].

### 6.1. Applications in Electronics

In general, the application field of self-assembly BC in electronics is in the modification of the electronic structure, which macroscopically is observed as the color variation of the thin film or in bulk. However, this change in the color may be employed as an indicator in sensors and electronic devices. Post-polymerization of poly(3-hexylthiophene) (P3HT)-*block*-Poly(phenyl isocyanide) (PPI) (poly(120-*b*-240)) modified with amine, promoted the self-assembly of blocks into supramolecular helical structures. This led to a positive Cotton effect at a wavelength of 364 nm, which was ascribed to the π-π* interactions of the P3HT segment and its immiscibility with the PPI domains, resulting in the exposition of PPI segments toward the exterior of the microstructure [124]. The polymerization of ultra-high molecular weight (UHMW) BC has been employed in the fabrication of micellar photonic crystals. Polystyrene as the first block, and poly(methyl methacrylate) (PMMA) or poly(2-vinylpyridine) (P2VP) as the second block, were employed. In order to tune the molecular separation between micelles, the BC (PS-*b*-PMMA or PS-*b*-P2VP) were dissolved in THF and polar solvents (acetone or methanol). Diluted, dried micelles observed by transmission electron microscopy (TEM) showed a spherical morphology and size in the range of 150–503 nm. The increase in the molecular weight of the BC influenced the UV-Vis spectra, observed as light reflection at wavelengths of 560 nm (green color), 635 nm (orange color), 670 nm (red color), and 765 nm (brown color). The self-assembly in bulk films showed a porous morphology composed of a continuous matrix of PS domains and PMMA or P2VP domains surrounding the pores. The nanometer-sized porous, acting as individual scattering centers, and the differences in the refractive index of the polymers, produced vivid structural colors on the films [125]. When a metallo-supramolecular P3HT polymerized in block with poly(ethylene oxide) (PEO) was formed with terpyridine-based ligands and ZnII ions, the self-assembled morphology in a mixture of CHCl_3_ and MeOH changed from spheres to fibers. This was observed as the color change from light yellow to purple as the content of CHCl_3_ was decreased, corresponding to the shift from 448 to 519 nm of wavelength in the UV-vis spectra [126]. P3HT-*block*-polystyrenesulfonic acid (P3HT-*b*-PSSA) in the form of flexible thin films may be employed in the fabrication of moisture-sensitive polymer-based flexible electronic devices. For this application, two properties of the BC were exploited, the adsorption capability of the sulfonated PS blocks, and the conducting characteristics of the P3HT blocks. Sulfonated PSSA blocks (91%) adsorbed a greater amount of relative humidity, whereas the conductivity increased with the adsorption of water [84]. Core-shell nanoparticles based on multiarmed poly(n-butylacrylate-*block*-polystyrene) (PBA-*b*-PS) starburst BCs showed an increment in the absorption of visible light at wavelength of 750 nm, as the number of the arms increased from 4 to 12 [89]. 

The formation of micelles in BCs occurs when one of the blocks is dissolved in a good solvent, whereas the other block partially dissolves or does not fully dissolve. Subsequently, the undissolved block will form the center of the micelle as a dense core, whereas the dissolved block will be arranged around the micelle in the form of a corona. These structures have been used to obtain nanostructured materials, and for drug delivery in medicine. Poly(ethylene glycol)-*block*-polystyrene (PEG-*b*-PS) diblock copolymer was partially sulfonated in the PS blocks. The dissolution in low pH water as a good solvent promoted the self-assembly in the form of micelles with size of 30–80 nm. TEM images revealed that the PS blocks were forming the core of the micelles, the PEG blocks formed the corona, whereas the sulfonated molecules were located in the interface. These nanostructures may be employed as templates for obtaining solid nanoparticles [127]. Block copolymers based on poly(methyl methacrylate)-*block*-3-sulfopropyl methacrylate (PMMA-*b*-PSPMA) showed a 10–20-nm micelle-like self-assembly morphology. The micelles are composed of a hydrophobic core of PMMA and a hydrophilic corona of PSPMA. By increasing the length of the PSPMA block, the repulsion between the micelles is greater, causing a greater separation between the particles without affecting the particle size [128]. The photoacoustic effect and the near-infrared (NIR) response of nanometer-sized micelles based on poly(ethylene glycol) (PEG) methyl ether-*block*-poly(DL lactide) (PEG2000–PDLLAx) have been used as indicators to target tumors in living animals [129]. The fluorescence property of amphiphilic brush copolymers (BBCPs) based on poly(norbornene), PEG, an alkyl bromide chain and cyanine dye, as NIR indicator, have been employed for imaging and selective drug delivery of natural anticancer compounds [130]. P3HT, hydrophilic poly(triethylene glycol allene) (PHA), and poly(phenyl isocyanide) (PPI) were polymerized to form triblock copolymers with tunable white-light emissions in different states such as solution, gel, and solid state [131]. A fluorescent block copolymer based on 1,8-naphthalimide-*block*-PEGMA300 was used as a biomarker in medical applications. The fluorescent intensity at a wavelength of 525 nm increased with the concentration of ethyl acetate-methanol mixed solvent. The self-assembly of BC in the form of spherical particles with a diameter of 120 nm, allows them to penetrate the cell wall through the process of endocytosis [22].

### 6.2. Applications in Lithography

The lithography process is employed to produce micro- and nanometric patterns, and involves the use of a protecting mask and the etching of a light-sensitive material. Thus, the selection of the material deposited on the substrate is of great importance to improve the properties of the final device. In this sense, BC-based inks play a very attractive role because their properties can be manipulated by modifying different aspects of the block polymer [50]. The ability of BC for self-assembly at the nanoscale level have triggered their use in high-resolution patterns employed as templates [6]. The BC structural hierarchy as a function of size involves monomer (0.5 nm), block copolymer (5 nm), micelles (50 nm), domains (100 nm), and crystals (>1 cm) [132]. Although micelles are the preferred structure for drug delivery applications [11,13,14] nano-domains are used in lithography. Sulfonated PS domains in P3HT-*b*-PS have shown improvement of the electrical properties of the π-conjugated backbone, resulting in a better solubility in polar solvents. This feature was exploited to produce a better dispersion of single-walled carbon nanotube (SWCNT) inks, which increased the electrical conductivity of the printed dispersion [133]. 

### 6.3. Applications in Photovoltaics

Factors such as low production costs and device versatility are attractive for the use of organic materials in photovoltaic cells. The use of BCs in this type of application allows for the selection of the monomers to promote their dissolution in different solvents, including water. Thus, when solidifying, nanodomains of donor and acceptor materials can be formed. Varying significant parameters in BCs such as molecular weight, solvent type, annealing temperature, and sulfonation degree, different morphologies may be obtained, including lamella, gyroid, hexagonally packed cylinders (HEX), and hexagonally perforated lamellae (HPL) [62,120].

The semiconducting polymer poly({4,8-bis[(2-ethylhexyl)oxy]benzo [1,2-b:4,5-b′]dithiophene-2,6-diyl}{3-fluoro-2-[(2-ethylhexyl)carbonyl]thieno[3,4-b]thiophenediyl}) (PTB7) was employed as the rod block, whereas poly-4-vinylpyridine (P4VP) was employed as coil in the self-assembly of the BC. The solidification from the solution produced Janus-like nanoparticles morphology with sulfur-enriched domains. This type of morphology promotes the interconnection between donor and acceptor domains [12]. The thermal decomposition of poly(neopentyl p-styrenesulfonate)-*block*-poly(p-styrenesulfonate) (PNSS-*b*-PSS) in P3HT-*b*-PSS BCs, employed as an interlayer in organic photovoltaics (OPVs), enhanced the power conversion efficiency of the photovoltaic device. In these type of devices, the nano-morphology is very relevant, because thermal decomposition increased the Flory−Huggins interaction parameter (χ) between P3HT and PSS blocks, which promoted microphase separation, and an increase in the electron density between the crystalline and amorphous domains [134]. Anyhow, thermal treatments do not always result in microphase separation. In the case of BCs comprising hydrophilic sulfonic segments, such as poly(styrene sulfonate) and an amorphous triphenyldiamine block, the annealing at temperatures above the Tg did not significantly alter the microstructure of the thin films. Thus, solvent annealing with DMF vapor promoted phase separation from nonordered micelles into vertically aligned domains with size of 21–30 nm [135]. In this sense, THF and water mixtures have been used to promote self-assembly of nanoscale cuboidal particles of amphiphilic diblock copolymers based on poly(acrylic acid)-*block*-poly(4-vinylbenzyl)-3-butyl imidazolium bis(trifluoromethylsulfonyl)imide (VBBI+Tf2N-), (PAA45-*b*-PIL23). Spherical morphologies were obtained at water/THF ratios of 1.2, 1.4, and 1.8, whereas cubesome particles were observed at ratios of 1.5 and 1.6. Microstructure evolution begins with the addition of water to the THF containing the BC, and the subsequent formation of multilayer vesicles within 1 h. After 1 day, vesicles coalesce into cubesome particles, and after 2 days, cube edges appear sharper [136]. In addition to the phase separation induced by differences in the polarity of solvents, another strategy is the increase in the molecular weight of the BC. Amphiphilic diblock copolymers based on poly(2-(methacrylolyoxy)ethyl ruthenocene-carboxylate) (PMAERu) and hydrophilic PEO, terminated with 2,2′-azobis(2-methylpropionitrile) (AIBN) as a radical initiator, were synthesized. As the length of the PMAERu block increased from 35 to 50 units, the morphology of the worm-like micelles evolved from a multi-connected worm-like morphology network to a single Y-connected morphology [137]. Thermal treatments and solubility in aqueous solution also have a combined effect on morphology transitions in thermoresponsive BCs. Poly(N,N-dimethylacrylamide)-*block*-poly(4-hydroxybutylacry-late-stat-diacetoneacrylamide) [PDMAC–P(HBA-*stat*-DAAM)] at low concentration (0.1% *w/w*) in water was subjected to temperatures of 1, 25, 50, and 70 °C. Corresponding morphology transitions from spheres, worms, vesicles, and lamellae were observed by TEM. Thermal energy increases the free volume of the polymer chains, which provokes the increase in the hydration of the hydrophilic segments of the BC and the subsequent evolution of the morphology [138]. Thermal removal of the protecting group by simple heating at 150 °C did not affect the self-assembly morphology, but just removed the *n*-butyl acrylate (*n*BA) from the ionic sulfonic groups [139].

### 6.4. Applications in Fuel Cells

For some applications, the use of BC with an amphiphilic character has beneficial results for the optimization performance of novel technologies such as fuel cells, in which the charged-neutral block copolymers exhibited higher proton conductivity than random ionic copolymers with commercially available membranes of Nafion [109,140]. According to the authors, it is explained that better ionic pathways are created by the interaction between polar groups (Figure 4). The proton exchange membrane fuel cell (PEMFC) is a highly efficient device that directly converts chemical energy into electrical energy, and the proton exchange membrane (PEM) is employed as a solid electrolyte, and as a physical barrier to transfer protons and to keep the fuel gas in the other side of the membrane [9,141]. The use of sulfonated BCs as membranes in PEMFC is very attractive because these materials upon solidification, present phase separation between the different blocks, which are chemically incompatible. This unique characteristic, which does not occur in polymer blends and random copolymers, promotes the formation of nanoscale domains that act as ion conducting channels [7,8]. In this sense, the blocks in polystyrene-*block*-(1,3-cyclo-hexadiene) (PS-*b*-PCHD) were sulfonated and fluorinated, respectively, to induce phase separation in spin-coated thin films from THF/water solvent. The characterization by AFM showed nanometric (50–80 nm) sulfonated PS islands surrounded by a net-like pattern of fluorinated PCHD. After annealing at 110 °C, phase separation increased, showing a worm-like morphology with a size of approximately 200 nm in the sulfonated PS domains [54]. Hexagonal boron nitride nanosheets (h-BN), high sulfonated poly (ether-ether-ketone) (sPEEK), and expanded polytetrafluoroethylene (ePTFE), were employed for low-humidification proton exchange membrane fuel cells. This membrane showed proton conductivity of 0.283 S/cm at 60 °C, and the lowest swelling degree [26]. In general, the proton conductivity of sulfonated BCs is directly related to the sulfonation degree, and in turn to the density of functional groups. Unfortunately, a high sulfonation degree also brings a high swelling degree, which leads to a dramatic decrease in the dimensional stability of the membrane [7,142]. BC membranes prepared from sulfonated polybenzophenone/poly(arylene ether) showed well-developed hydrophilic/hydrophobic phase separation, where interconnected hydrophilic domains containing sulfonate groups were about 5–10 nm wide. This membrane presented a proton conductivity (0.45 S/cm) 3.7 times higher than Nafion. This behavior was attributed to the highly acidic sulfonic acid groups, due to the presence of carbonyl groups attached on the same phenylene rings [10]. Sulfonated PMMA-*b*-PS cast as membranes showed an increase in proton conductivity with the sulfonation degree and with testing temperature, from 0.0089 to 0.015 S/cm, and 0.0092 to 0.026 S/cm for 27 and 40% of sulfonation, respectively. The increase in conductivity with the degree of sulfonation is due to the higher density of proton charge carriers available in the hydrophilic domains, while the increase with temperature is due to the decrease in the energy barrier for proton transfer [89]. Similar BCs synthesized by ATRP containing 7% of PMMA, showed a higher proton conductivity (0.151 S/cm) at a sulfonation degree of 24.6% [52]. Triblock copolymers based on poly(styrene-*block*-vinylidene fluoride-*block*-styrene) (PS-*b*-PVDF-*b*-PS) were sulfonated in the PS blocks with varied degrees. Self-assembly morphology evolved from lamellar to a disrupted morphology at sulfonation degree of 13%, and to a lamellar morphology with large-scale phase separation at sulfonation above 23%. Corresponding proton conductivities up to 0.09 S/cm were observed at water uptake of 95% [64]. Diblock copolymers of sulfonated poly([vinylidene difluoride-*co*-hexafluoropropylene]-*block*-styrene) [P(VDF-*co*-HFP)-*b*-SPS] were employed to demonstrate that the sulfonation degree has a higher impact on the proton conductivity than the self-assembly microstructure. For instance, the ion exchange capacity (IEC) increased from 0.31 to 0.73 mmol/g with a sulfonation degree of 16 and 41%, respectively. This was attributed to the increase in length from 6 to 20 nm of the ion clusters containing the sulfonated PS blocks [53]. Triblock copolymer membranes, based on polystyrene-*block*-poly(4-vinylpyridine)-*block*-polystyrene (PS–P4VP–PS) with different acid doping level (ADL), were employed to test the influence on proton conductivity. When sulfuric acid (Sa) was employed as the doping agent, as the ADL increased from 0 to 4.6 mol/mol, the Tg decreased from 150 °C to −85 °C, respectively. Conversely, the proton conductivity increased by three orders of magnitude, from 0.0001 to 0.5 S/cm. This behavior was explained in terms of the ease of dissociation of the Sa within the triblock copolymer in the range of low to near room temperatures [87]. Yan et al. induced the formation of gyroid morphologies, by the formation of ionic aggregates, in a series of precisely segmented polyethylene-like materials, containing sulfonate groups (PES23) using different counterions such as Li^+^, Na^+^, Cs^+^, or NBu^4+^. The self-assembly process and hence the obtained final morphology is dependent on the cation. For example, as observed in Figure 5, PES23, having as counterions Li^+^, Na^+^, and Cs^+^ cations, exhibited a transformation upon melting into a gyroid morphology, whereas the gyroidal ionic aggregates further evolve into hexagonal symmetry as the temperature increases. In addition, it was also observed that the ionic conductivity (IC) depends on the morphology adopted by the formation of ionic aggregates, being 3D-interconnected gyroid morphology of PES23Li, which exhibited the higher IC, in comparison with an isotropic layered or hexagonal symmetry. According to the authors, the symmetry of the ionic aggregate morphologies plays a critical role in ion transport, and the size of the counterions may be employed to control the self-assembly of the obtained nanostructure [143].

Additionally, to the increase the protonic conductivity with the sulfonation degree, the block polymers present a greater stability to swelling with the adsorption of water. Sulfonated poly(styrene–isobutylene–styrene-*block*-vinyl phosphonic acid) (SIBS-*b*-PVPA) showed a swelling value of 1.3% at a water content of 95%, whereas the non-sulfonated (SIBS-*b*-PVPA) type showed values of 2.5 and 15%, respectively. Corresponding proton conductivities were 0.055 and 0.014 S/cm. The structural behavior was explained in terms of the ionic interactions between PO_3_H_2_ and SO_3_H that formed crosslinking through sulfonate–phosphonium complexes [56].

## 7. Conclusions

In the present work, the recent literature published in the last two decades related with the self-assembly process of charged-neutral block copolymers was reviewed. The work gives an overview of the available living controlled polymerization methods, which are commonly employed for the synthesis of BCs. Additionally, it was analyzed by different methods for the obtainment of an amphiphilic character introduced through the sulfonation of one of the segments. The current literature mainly deals with polymers that include aromatic rings in the backbone chain because the sulfonation readily proceeds through an electrophilic aromatic substitution mechanism. We placed emphasis in published works that presented evidence of the morphological changes adopted upon the introduction of the polar groups. The authors describe that the self-assembly process could be related to different factors; among them, it is mentioned that the presence of different types of electrostatic interactions inter and intra-chain, dipole–dipole, hydrogen bonds, among others. In addition, it has also been observed that the size of the counterions plays an important role, and the type of treatment employed in order to reach the equilibrium morphologies, including thermal, solvent vapor annealings, and the application of electric fields, which are the most used for this purpose. Additionally, the authors agree that the interaction parameter, which describes the repulsive energies between the component blocks, is also modified because the chemical dissimilarity of the constituent blocks increases for a higher and more positive value of χ is observed.

The sulfonation reaction represents a tool by which to induce the formation of desired morphologies that could not be accessed for uncharged polymers, resulting in an enhancement of the mechanical, optical, and electrical properties of the materials employed for the development of emerging technologies.

## Figures and Tables

**Figure 1 polymers-14-05081-f001:**
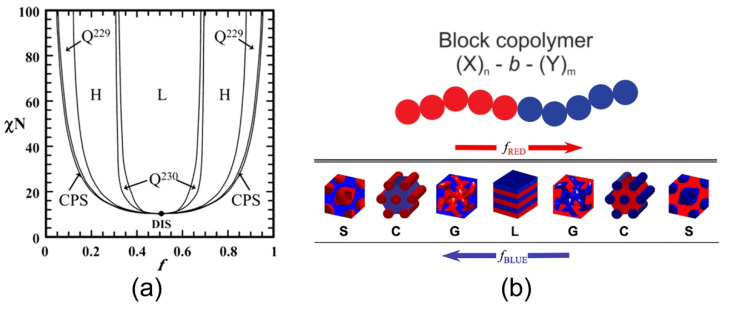
(**a**) Theoretical phase diagram of diblock copolymer melts calculated by using the SCFT and schematic representation of different morphologies. Reprinted with permission from reference [35] Copyright 2006 American Chemical Society. *f*, volume fraction of one block; χ, Flory–Huggins interaction parameter; N, degree of polymerization; L, lamellae; H, hexagonally packed cylinders; Q_230_, double-gyroid phase; Q_229_, body centered spheres; CPS, closed-packed spheres; and DIS, disordered. (**b**) Equilibrium-ordered state transitions in diblock copolymers determined by relative volume fraction f. Spherical (S), cylindrical (C), bicontinuous gyroidal (G), and lamellar (L)microdomain organization nanostructures of a linear AB diblock copolymers. Adapted with permission from reference [36].

**Figure 2 polymers-14-05081-f002:**
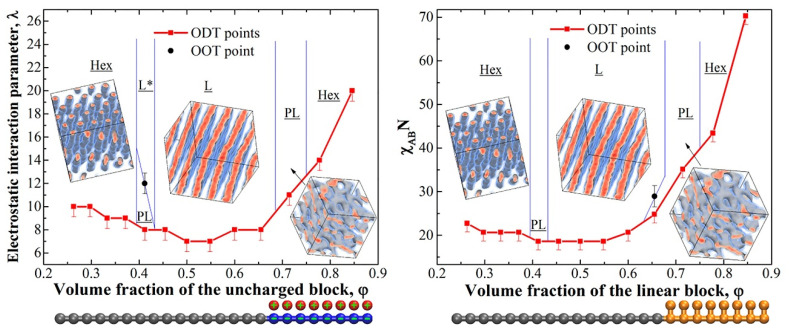
Phase diagrams of diblock-copolymers with one charged block (**left**) and uncharged diblock-copolymers for the main (linear) chain length of N = 24 (**right**). Reproduced from reference [119].

**Figure 3 polymers-14-05081-f003:**
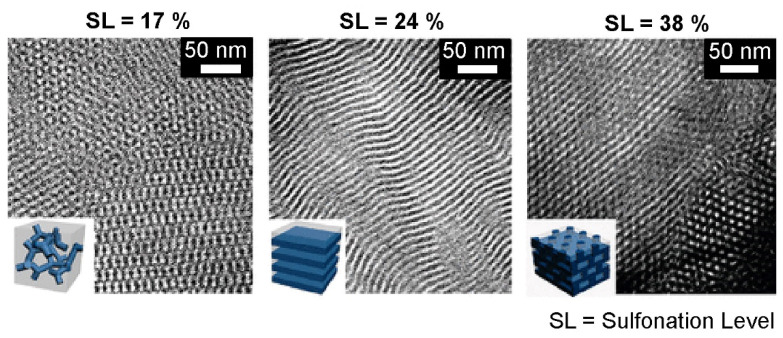
Phase behavior of poly(styrenesulfonate-methylbutylene) (PSS-PMB) block copolymers. Evolution of the morphological changes upon increasing sulfonation level from 17 to 38%. Reprinted with permission from reference [120]. Copyright 2008, American Chemical Society.

**Figure 4 polymers-14-05081-f004:**
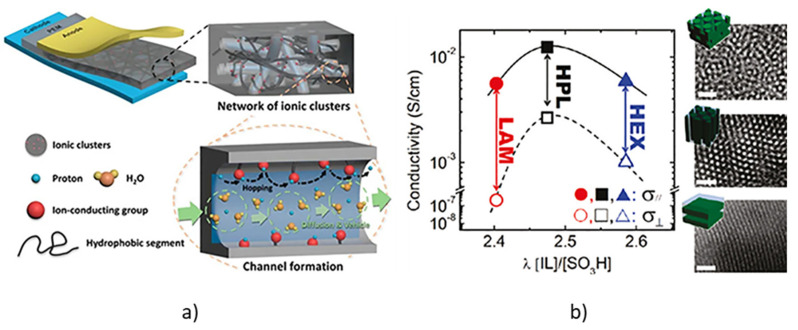
(**a**) Formation of ionic channels induced by interaction of polar groups that favors the proton exchange capacity in fuel cells. (**b**) Different morphologies adopted by the relation of ionic liquids and sulfonic acid groups and its corresponding ionic conductivity. Adapted with permission from reference [76]. Copyright 2021, American Chemical Society.

**Figure 5 polymers-14-05081-f005:**
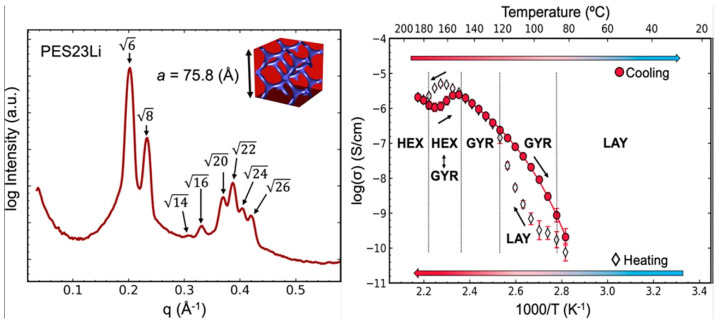
Small-angle X-ray scattering profile (SAXS) of polyethylene sulfonated at 150 °C exhibits the Ia3d cubical gyroid phase having a Li^+^ as counterion. And evaluation of the its ionic conductivities upon cooling from 190 to 25 °C, ∼1 °C/min followed by a second heating of 25 to 190 °C, ∼1 °C/min. Adapted with permission from reference [143]. Copyright 2021, American Chemical Society.

**Table 1 polymers-14-05081-t001:** List of recent works published in the last decade that employed one sulfonated monomer to the obtention of the BC.

Polymerization Technique	BC System	Procedure for Obtaining of ABC	Applications	Reference
RAFT	Poly((sodium2-acrylamido-2-methylpropane-sulfonate)-b-ethylene glycol)	First, 2-(acrylamido)-2 methyl propanesulfonic (AMS) was polymerized by NMP. Subsequently extended with a macro RAFT agent of 2(methacryloyloxy)ethyl phosphorylcholine (MPC) for diblock copolymers and poly(ethyleneglycol)-based bifunctional chain transfer agent for the synthesis of tri-blocks PAMPS-b-PEG-b-PAMPS.	Medical science and drugs development. The diblock copolymer arquitectures based on PAMPS works more effectively for its anticoagulant activity.	Kalaska et al. 2018 [69]
ATRP	Poly(ethylene glycol monomethacrylate-b sodium 4-styrenesulfonate)	A membrane of chloromethylated of poly(ether imide) was used as a surface active initiators. The polymerization of PEG or PNaStS was carried out on the membrane and the block copolymerization was completed after reactivation of the preserved dormant chain ends with 2,2 –byridine, copper(I) chloride and copperchloride.	Membranes for ultra and microfiltration and protein adsorption.	Li et al. 2015
ATRP	Poly(sodium styrene sulfonate-b-methyl methacrylate)	First, PSSNa macroinitiator was synthesized and then used used as initiator methyl-4-(bromomethyl)benzoate, copper(I) bromide (CuBr) as catalyzer, the ligands N,N,N′,N″,N‴ pentamethyldiethylenetriamine and 2,20-bipyridine, the surfactant hexadecyltrimethyl ammonium bromide and the solvents methanol and DMF.	Medical and food science for antimicrobial materials.	Oikonomou et al. 2011 [70]

**Table 2 polymers-14-05081-t002:** Recent literature published employing post sulfonation reaction into the BC.

Polymerization Technique	BC System	Obtention of ABC Character	Applications	Reference
Anionic polymerization	polystyrene-*b* poly(sulfonated isoprene co-isoprene)	Sulfur trioxide/1,4-dioxane complex was used as AS and added dropwise to a solution of the BC (T = 25 °C, t_rxn_ = 4 h) neutralization was with NaOH and methanol to stop the reaction. The sulfonated BC was recovered by dyalisis.	Not mentioned	Politakos et al. 2021 [62]
Anionic polymerization	Poly(4-(4sulfobutyloxy) styrene-*b*-(4-(n-butoxystyrene)	AS was a solution of methanesulfonic acid in DMSO (T = 80 °C, t_rxn_ = 36 h).	Proton exchange membranes	Sheng et al. 2013 [71]
Not synthesized	Crosslinked sulfonated poly(styrene-*b*-butadiene-*b*-styrene) (SBS)	AS was acetyl sulfate freshly prepared from acetic anhydride in DCE and sulfuric acid at T = 0 °C then added to a swollen film of crosslinked SBS (T = 50 °C, t_rxn_ = 30 min).	Proton exchange membranes	Won et al. 2003 [72]
Anionic polymerization	Sulfonated poly(1,3-cyclohexadiene-*b*-ethylene glycol)	1,3- cyclohexadiene was polymerized and crosslinked previous to the copolymerization reaction with ethylene oxide. The BC was then sulfonated with a solution of chlorosulfonic acid in dichloroethane was used as AS (T = 25 °C, trxn = 1 h).	Proto exchange membranes	Deng et al. 2015 [73]
ATRP	Sulfonated poly(vinylidene difluoride-cohexafluoropropylene-*b*-styrene)	Trichloromethyl-terminated fluoropolymers of vinylidene difluoride and hexafluoropropylene were prepared by emulsion in chloroform as the chain transfer agent. The resulting CCl_3_- terminated were used as macroinitiators to extend the polystyrene block. The AS was acetyl sulfate.	Proton exchange membranes	Tsang, Shi and Holdcrof 2011 [53]
Anionic polymerization	poly(styrenesulfonate-*b*- tert-butylstyrene)	The BC were prepared by polymerization first of the tert-butyl styrene by using sec-butyllithium as initiator followed by the addition of styrene to the living chains, then a solution of sulfur trioxide in dichloroetane was employed as AS (T = 0 °C).	Not mentioned	Yang et al. 2001 [74]
Not synthesized	Sulfonated poly[(styrene-*b*- (ethylene-alt-propylene)]	Sulfonated with acetyl sulfate in various chlorinated solvents (CHCl_3_ or CH_2_Cl_2_ or 1,2-dichloroethane) prepared by reaction of sulfuric acid and acetic anhydride. (T = 50 °C, t_rxn_ = 3 h).	Proton exchange membranes	Gromadzki et al. 2006 [75]
Anionic polymerization	Poly(styrenesulfonate-*b*-methylbutylene)	First, the polymerization of styrene and isoprene was carried out and followed by selective hydrogenation of the polydiene. A solution of freshly prepared acetyl sulfate was used. (T = 40 °C, t_rxn_ = 1,4 and 25 h).	Proton exchange membranes	Kim, Kim and Park 2011 [76]
	Sulfonated poly(styrene-*b*-ethylene/butylene-*b*-styrene)	The BC was dissolved in DCE and the AS (acetyl sulfate) was added to the solution. (T = 50–53 °C, t_rxn_ = 2 h).	Templates for magnetic nanocomposites	Peddini et al. 2015 [77]
Anionic polymerization	Poly(styrene-*b*-sulfonated hydroxystyrene)	The sulfonic acid groups were grafted onto the PHS segment by reacting the BC with potassium hydride and 1,3-propanesultone in anhydrous THF (T = 60 °C, t_rxn_ = 24 h).	Proton exchange membranes	Lee et al. 2011 [78]
Cationic polymerization	Sulfonated poly(styrene-*b*-isobutylene-*b*-styrene)	BCs were lightly sulfonated with acetyl sulfate in refluxing methylene chloride.	Not mentioned	Storey et al. 2000 [79]
Anionic polymerization	Poly(isoprene-*b*-sulfonated styrene)	The BCs were fluorinated by reacting the double bonds of the PI with difluorocarbene and were then sulfonated. (sulfonation conditions not reported).	Polymer electrolyte membranes	Sodeye et al. 2011 [80]
Anionic polymerization	poly(diethylsilacyclobutane -*b*- methyl methacrylate)	Sulfonation of BCs employed 1,3-Propane Sultone as AS. The BC was dissolved in a sodium hydroxide aqueous solution and then an excess of 1,3- propane sultone was added and neutralized by the addition of sodium hydroxide.	Medical science as drug deliveries and biocompatibilizers.	Matsuoka et al. 2003 [81]
Not synthesized	Poly(styrene-*b*-ethylene-co-propylene)	Sulfonation was carried out with acetyl sulfate in 1,2-dichloroethane. It was prepared by by the reaction of concentrated sulfuric acid with 30 mol% excess of acetic anhydride.	Compatibilizers	Zhan et al. 2000 [82]
Anionic polymerization	Poly(styrene-b-isoprene) and Poly(styrene-b-dimethylsiloxane)	Blocks were hydrogenated to prevent degradation and to favoring the selective sulfonation of polystyrene. Acetyl sulfate prepared in acetic anhydride and sulfuric acid. (T = 40 °C and trx = 4 h).	Water purification and ion exchange membranes.	Hernández et al. 2013 [83]
Anionic polymerization	Poly(isoprene fluorinated -*b*- sulfonated styrene)	First, the PS/PI BC was synthesized followed by a fluorination procedure. The sulfonation was carried out with acetic anhydride and sulfuric acid (T = 25 °C, t_rxn_ = 24 h).	Molecular electronics, photovoltaic, and fuel-cell membranes	Goswami et al. 2010 [84]
Click chemistry and ATRP	Poly(3-hexylthiophene)-b-styrenesulfonic acid)	The obtention of the BC started with synthesis of bromobenzyl end-functionalized P3HT as ATRP macro initiator, followed by the polymerization of styrene and finally, the sulfonation of PS block carried out by solution of phosphorous pentoxide in concentrated sulphuric acid (T = 40 °C, trxn = 30 min).	Humidity sensors	Khawas et al. 2019 [68]
Anionic polymerization	Poly(styrene-*b*-isobutylene-*b*-styrene)	The BC was dissolved in dichloroethane and the sulfonation agent was (t_rxn_ = 2 h).	Membranes for water vapor-breathable films	Mountz et al. 2005 [85]
RAFT	Poly(methyl methacrylate-*b*-styrene sulfonate)	Methyl methacrylate macro RAFT agents were employed to reactive the copolymerization reaction with styrene monomer. Subsequently, the sulfonation of the PS segments was carried out by using freshly prepared acetyl sulfate (T = 40 °C, t_rxn_= 5 h).	Proton exchange membranes	Piñón et al. 2019 [33]
RAFT	Poly(styrene-*b*-4 vinylpyridine)	Free pyridine unit were subjected to reacted with propane sultone to give catalyst precursors 3–6 which were followed by the acidification by trifluoromethanesulfonic acid (AS)to produce the corresponding acidic ionic liquid.	Catalysis for the synthesis of biodisel	Jiang et al. 2020 [86]
RAFT	Poly(styrene-*b*-4-vinylpyridine-*b*-styrene)	First, styrene was polymerized by a bifunctional RAFT agent and copolymerized and then extended with the styrene monomer. Then, acid-swollen membranes were prepared by dissolving the obtained BCs in pyridine. Finally, sulfonation procede through by a 98 wt% aqueous solution of sulfuric acid.	Electrolyte membranes	Kajita et al. 2021 [87]
RAFT	Poly (methyl methacrylate-b-styrenesulfonate)	The synthesis of the BC was carried out by seeded emulsion polymerization with 1,1-diphenylethylene (DPE) as a chain transfer agent. The sulfonation employed acetyl sulfate (T = 40 °C, t_rxn_ = 75 min and 130 min).	Proton exchange membranes	Wang et al. 2013 [88]
ATRP	Poly(styrenesulfonate-*b*-methyl methacrylate)	The BC were successfully converted to their ionomers by sulfonation using acetyl sulfate as sulfonating agent (T = 30 °C, t_rxn_ = 45 min).	Proton exchange membranes	Erdogan et al. 2009 [52]
ATRP	Poly(n-butyl acrylate-*b*-polystyrene sulfonate)	The outer PS shell of the star copolymer was converted into hydrophilic poly(p-styrenesulfonate) with acetyl sulfate (t_rxn_ = 24 h). Finally, the oxidative propagation of 3,4-ethylenedioxythiophene on the PSS chains was carried out by counterion-induced polymerization to produce a stable aqueous dispersion.	Electrically conductive core-shell nanoparticles	Chu et al. 2008 [89]

## Data Availability

Not applicable.

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
