# Peer review of "Sulfonated Block Copolymers: Synthesis, Chemical Modification, Self-Assembly Morphologies, and Recent Applications"

_polymers, 2022, doi:10.3390/polym14235081_

Round 1

Reviewer 1 Report

This manuscript reviews sulfonated block-copolymers and focuses on their synthesis, modification, self-assembly, and applications. The authors report that the review summarizes the relevant publications of the last two decades. The "References" section confirms that the survey of the scientific literature is adequate and mostly represented by up-to-date papers published in high quality journals.

The manuscript is within the scope of "Polymers" journal and its special issue "Phase Transitions and Structures in Polymer Science".

The authors discuss sulfonated amphiphilic block-copolymers as structural units of self-assembled nanosystems. The structure-morphology-property relationships summarized in this review are interesting in terms of potential applications that such polymer materials can find in industry and medicine. The phase behavior of modified copolymers is also discussed in detail.

The manuscript will be of interest for the audience of "Polymers" and specifically its special issue "Phase Transitions and Structures in Polymer Science". I recommend, however, its thorough revision before it can be approved for publication.

1. The manuscript requires extensive editing of English. Language style and grammar throughout the text reduce its readability and sometimes make the scientific message unclear.

2. The text is also hard to read because of large paragraphs (the one in page 5, for example, occupies this entire page).

3. The page layout is landscape. In the portrait layout mode, the paragraphs will be even larger and harder to read.

3. There is a link to Fig. 1 in the Introduction. Will it be more appropriate to discuss the phase diagram in Fig. 1 after the Introduction? Otherwise, please clarify how Fig. 1 contributes to stating the aim of this Review.

4. In Section 4, the authors focus on structural and morphological changes induced by sulfonation. Both the nanoscale organization and phase behavior of block-copolymers are discussed. While the phase behavior of polymers is among the key priorities of this Special Issue ("Phase Transitions and Structures in Polymer Science"), nanoscale structures are important for practical applications of polymer systems. Is it possible to introduce sections 4.1 and 4.2 to this Review to separately discuss the phase behavior and nanoscale morphologies of sulfonated block-copolymers?

5. An additional figure, which will demonstrate examples of self-organized nanoscale block-copolymer units, will also be helpful for this review, if available.

6. Lines 225-227 and 229-231: please add relevant references.

7. Line 275: It seems to be helpful to provide these phase diagrams, is they are available.

8. Line 320: the authors mention drug delivery applications of block-copolymers. If applicable, this review may benefit from a more detailed discussion on how sulfonated amphiphilic block copolymers contribute to developing drug delivery systems.

Author Response

The authors sincerely thank the reviewers for the valuable comments and changes suggested in the manuscript under review. All of these were considered and answered individually. We hope that these answers meet your expectations. As authors, we can say that the work was modified extensively, obtaining a more readability version and better discussed work. Please find in the following sections, the answers for each of your comments and questions. The changes made in the text were highlighted in red font in the present document, the formal manuscript revised version (Microsoft Word file), and in the PDF file. The answers for the reviewer not included in the text, are in black font.

Reviewer 2 Report

The review paper by Zaragoza-Contreras et al. offers an actual report on synthesis, chemical modification, and morphology of sulfonated block copolymers used for a number of applications. The paper is composed in a logic way and comprehensively described. It is scientifically sound and well focused on the topic. In general, the list of references is complete. I may however recommend adding several references to the point covering alteration of BC morphologies via introduction of additives by dipole-dipole, van der Waals, and hydrogen bonds (line 59).

There is a number of typos and awkward sentences that have to be corrected to avoid misleading. I marked them in the file attached to my review.

In particular, I recommend using uniformly the parameter of "sulfonation degree" instead of different terms, such as "sulfonation values" (line 154)  or "sulfonation efficiency" (line 199), or simply "a sulfonation of 70%" (line 173).

Also: what is MDEA (line 241)?

I support publication of the review. I believe it may be published after minor revision to address the corrections mentioned in these comments and marked as comments in the attached pdf file is done.

Author Response

(The authors gave the same response as above.)

Round 2

Reviewer 1 Report

The authors addressed my major revision concerns and added detailed clarifications of the previous review comments.

Comparison of the two versions of submitted manuscript files reveals, however, that it still requires language editing and some previous grammar and/or style imperfections have not been addressed, for example:

The sulfonation reactions represents... (Line 13)

The use of these techniques not only provide (Line 177)

The amphiphilic nature in a BC can be introduced by two strategies (Line 189) 

The residual language imperfections in this text are not limited by these several examples. The readability of the revised manuscript is, however, much better as compared with the original sumbitted paper.

The authors are recommended to carefully re-read the revised manuscript to eliminate the residual language concerns.

Author Response

The authors sincerely thank the reviewers for the valuable comments and changes suggested in the manuscript under review. All of these were considered and answered individually. We hope that these answers meet your expectations.  The changes made in the text were highlighted with red font in the present document, the formal manuscript revised version (Microsoft Word file), and in the PDF file. 
